# The Old Man and the Meat: On Gender Differences in Meat Consumption across Stages of Human Life

**DOI:** 10.3390/foods10112809

**Published:** 2021-11-15

**Authors:** Christian Ritzel, Stefan Mann

**Affiliations:** 1Agroscope, Economic Modelling and Policy Analysis, 8356 Ettenhausen, Switzerland; christian.ritzel@agroscope.admin.ch; 2Agroscope, Socioeconomics, 8356 Ettenhausen, Switzerland

**Keywords:** meat consumption, gender identity, masculinity

## Abstract

It is not a radically new insight that men eat more meat than women do. However, one piece of the puzzle was previously missing: the development of a gender bias in total and red meat consumption across stages of human life. To identify the gender bias across stages of human life, we apply a multiple-group regression across seven age classes. Data for the empirical analysis stem from the US National Health and Nutrition Examination Survey. Regression results reveal that gender differences in meat consumption start only after the age of four and then move in some parallel with the development of biological differences, reaching a maximum between 51 and 65 years. The effect of both household income and education on meat consumption is negative and per-capita consumption of meat rises with household size.

## 1. Introduction

Meat consumption provides multiple societal challenges. Fundamentally altering the ratio between plant-based and animal-based food in our diets may be the only lever to bring agricultural production to some level of sustainability with respect to greenhouse gas emissions [1]. Reducing meat consumption would have multiple benefits for public health [2] and would solve some of the problems that ethicists raise when they question the practice of killing animals for human consumption [3,4].

Due to the grave environmental, health-related, and ethical challenges of meat consumption, understanding and altering the split between crop-based and animal-based calories may be one of the main factors in the transformation towards a sustainable food system. However, this necessitates a thorough understanding of the factors influencing the demand for meat. The most stable pattern in this respect is likely the difference between genders: men eat considerably more meat than women do [5] and much more red and processed meat, which has an even greater environmental footprint [6]. Understanding the underlying reasons for gender differences in meat consumption, therefore, is a good starting point for determining the underlying reasons for meat demand.

This paper uses age as a potentially important intermediate variable. While age itself usually is not an important predictor of the amount of meat consumed, we are interested in whether the gender bias regarding meat consumption remains stable during the different phases of life. Filling this research gap may give hints about cultural and biological influences on the gender differences in meat consumption, as will be outlined in detail below. While we display the state of knowledge about the gender gap in meat consumption in Section 2, we develop our hypotheses on the influence of age on the gender gap in Section 3. The data and method for testing these hypotheses are presented in Section 4. The results are discussed in Section 5, and Section 6 concludes the paper.

## 2. Meat Consumption and Masculinity

The notion that food items can signal masculinity or femininity enjoys great acceptance among sociologists. Linden flower tea [7], Caprese salad [8], or sushi [9], for example, have been described as feminine food. Conversely, masculine food items may include chunky fudge cake ice cream [7], beer [10], meat in general [11,12,13], and red meat in particular [14,15,16,17] receive the most frequent mentions by far as typical food options for men.

These societal connotations are more or less in accordance with reality. The fact that men eat more meat than women do is consistent for a large number of countries, and even the relation between male and female consumption amounts is similar, whereas, for example, the impact of education on meat consumption differs considerably, between countries [18].

While it is easy and common to exploit such gender-specific differences in advertising campaigns [19,20,21], it is much less easy to find the underlying reasons for them. Do biological differences between men and women, such as testosterone levels [22], have an impact on food preferences? Or should we lean toward Buerkle’s [23] suggestion that beef consumption among men is just a socially constructed way to fortify masculinity?

Some authors have shed light on this question by using personal values as a mediating variable. Mertens et al. [24], for example, find that men who support Machiavellianism and believe in the importance of power eat more meat than others. Similarly, Hayley et al. [25] identify the belief in power and the values of security and conformity as mediating variables between men and meat consumption. For women, universalism is the value that predicts little meat consumption. This information may make it understandable that exposure to the meat-animal condition in an experiment by Dowsett et al. [26] contributed to less meat consumption for women and more for men.

A variable that may have been neglected to understand the gender difference in meat consumption is age. How gender roles evolve and what impact this may have on meat consumption will be explored subsequently.

## 3. Age as a Mediating Variable in the Gender Bias

If the possibility exists that the difference in meat preference between genders is biologically determined or co-determined, it is important to briefly recall the evolution of gender differences over the course of life. As is well known, we only have our primary sexual characteristics once we are born. The first secondary sexual characteristics, such as the second-to-fourth finger length ratio, develop around at the age of 4 [27], which is also when the individual’s sense of being male, or female develops [28]. In the following years, most other secondary sex characteristics are formed, most prominently briefly before and during puberty [29]. If biology forms the differences in meat consumption between genders, these differences would emerge during childhood and adolescence and remain relatively constant after this.

The pattern of gender differences in meat consumption would probably look different if it was only culturally induced. Although parents may react to food avoidance by their children with slight changes [30], they strongly determine their children’s diet during childhood [31,32]. These early influences are shaped to geographical patterns [33] and must be considered as cultural induction. However, parents are always the main actors to transmit cultural norms to the next generation. Pink and blue colors in the clothing of US girls and boys from the fourth month onwards [34] serve as a case in point for a cultural norm that parents play a key role in maintaining. If the gender differences in meat consumption were due to the fact that parents, in one way or another, encourage their sons or discourage their daughters to eat meat, gender differences would form quickly and probably remain stable over the course of life. Such a culturally induced, constant gender gap over the course of life is our underlying hypothesis for this study.

This makes it conceptually promising to take a closer look at the evolvement of gender differences over different age classes. For the definition of age classes, we refer to Armstrong’s [35] 12 stages of human life. Armstrong’s 12 stages of human life, however, include the stages pre-birth (Stage 1), birth (Stage 2), and death (Stage 12), which are obviously not relevant for our analysis. Accordingly, Table 1 shows Stages 3–11 of human life and the corresponding age classes 1–7 used for the empirical analysis.

We slightly modify Armstrong’s 12 stages of human life to achieve equally balanced age classes regarding the number of observations. According to Bradley and Zucker [28], individuals develop a sense of being male or female at age 4. Age Class 1, therefore, refers to infancy (0–4 years). Stages 4 to 6 (early, middle, and late childhood) are combined into Age Class 2, childhood (ages 5–11). Stage 7 corresponds to Age Class 3, adolescence (ages 12–20). Stage 8 corresponds to Age Class 4, early adulthood (ages 21–35). Stage 9 corresponds to Age Class 5, midlife (ages 36–50). Stage 10 is spilt into Age Class 6, mature adulthood (ages 51–65 years), and Age Class 7, late adulthood (ages 65–80 years). Individuals 80 and over are top-coded in data used at 80 years of age.

## 4. Data and Method

### 4.1. Data

For our empirical analysis, we use data from the National Health and Nutrition Examination Survey (NHANES) of the National Center for Health Statistics (NCHS) of the United States [36]. The main aim of the NHANES is to collect the health and nutrition status of adults and children living in the United States. For this purpose, each year approximately 7000 participants are randomly selected. The NHANES comprises the following two parts: first, an interview containing questions on socio-demographic characteristics, individual nutrition behavior, and health; second, a physical examination consisting of medical, dental, and physiological measurements as well as laboratory tests administered by medical personnel. The sample is selected to represent the US population with respect to all ages. People 60 and over, African Americans, Asians, and Hispanics are oversampled in the NHANES [37].

Table 2 presents the description and summary statistics of our data. For the empirical analysis, we compute following three new variables: first, the share of total meat consumption in relation to total food consumption (excl. beverages); second, the share of red meat consumption in relation to total food consumption (excl. beverages); and third, the share of at-home consumption (incl. beverages) in relation to food and beverages consumed outside the home and at home. To calculate the share of total meat consumption, we include all meat types of the official food coding scheme (food coding scheme chapter 2 “Meat, Poultry, Fish, and Mixtures”). Meat types considered “red meat” can be found in Table A1 in Appendix A.

Regarding socio-demographic characteristics, some of the variables are directly related to the respondent (age and sex), whereas others (education, income, and marital status) relate to the household reference person or the household, respectively. Especially for respondents under 20 years of age, education, income, and marital status are not available. Therefore, we consider variables referring to the household reference person.

For the empirical analysis, we use data from five survey waves, covering the years 2007 to 2016. After data cleaning (dropping missing values), in total, our dataset consists of 41,262 observations. Even though the data have a time dimension through combining five survey waves, we do not observe the same respondents over time. Consequently, the data used cannot be characterized as panel data.

### 4.2. Method: Multiple-Group Regression

#### 4.2.1. Model Specification

To estimate the effect of gender (and other explanatory variables) on the share of meat consumption across seven age classes, we apply a multiple-group regression analysis. The specification of the model to be estimated is as follows:$$Y_{\mathrm{it}\;}{=\beta_{0}+\beta }_{1}{\mathrm{GEN}}_{\mathrm{it}}{+\beta }_{2}{\mathrm{EDU}}_{\mathrm{it}}{+\beta }_{3}{\mathrm{HHI}}_{\mathrm{it}}{+\beta }_{4}{\mathrm{HHS}}_{\mathrm{it}}{+\beta }_{5}{\mathrm{MAR}}_{\mathrm{it}}{+\beta }_{6}{\mathrm{SHC}}_{\mathrm{it}}{+A}_{t}{+A}_{e}{+\varepsilon }_{\mathrm{it}}\;$$
where *Y_it_* represents the share of meat consumption in relation to total food consumed (in %) of respondent *i* at time (wave) *t*. *β*_0_ represents the intercept, and *β*_1_ to *β*_6_ depict the parameters of interest to be estimated. In addition to socio-demographic characteristics, we include time (wave) fixed effects (FE) *A_t_* and ethnicity fixed effects *A_e_* in the regression analysis. In general, including fixed effects allows for controlling for unobserved time-invariant factors. In this context, especially a respondent’s ethnic or cultural background is assumed to influence nutritional behavior [38,39]. *ε_it_* denotes the error term for unobserved characteristics of respondent *i* at time *t*, which is assumed to fulfil the zero conditional mean assumption.

The equation is separately estimated for (1) share of total meat consumption (in %) and (2) share of red meat consumption (in %). Considering the share of meat consumed in relation to total food consumption instead of the absolute amount of meat consumed is more meaningful. For instance, respondents whose diet rests to a large share upon meat (e.g., 90% or even 100%) can be made visible by computing the share. As a robustness check, we estimate the equation separately for the amount of total meat consumed (in grams) and the amount of red meat consumed (in grams). Results with amounts of meat consumed in grams as the dependent variable can be found in Table A2 in Appendix A.

*GEN* is a binary variable that takes the value of one if a respondent is male and zero if a respondent is female. *EDU* is an ordinal-scaled variable depicting the educational level of the household reference person. The ordinal-scaled variable household income is represented by *HHI*. *HHS* depicts the count variable household size. *MAR* is a binary variable taking the value of one if the household reference person is married and zero otherwise. *SHC* represents the share of home food consumption (incl. beverages) in relation to food consumption (incl. beverages) at home and outside the home.

#### 4.2.2. Choice of Estimation Technique

The two dependent variables are fractional response variables, which need to be bounded between zero and one (0 ≤ Y ≤ 1). Therefore, estimating the model outlined in the previous chapter by means of ordinary least squares (OLS) regression is not feasible because this method does not guarantee that predicted values lie within the boundaries of 0 and 1. Figure 1 indicates that data are strongly skewed to the right and zero values are frequent; thus, OLS regression is clearly not appropriate. Against this background, Papke and Wooldridge [40] suggest estimating a generalized linear model (glm). In Stata, this can be performed by using the glm command. The glm command allows a flexible specification of the distributional family of the dependent variable (e.g., Poisson or negative binomial) and the transformation function (e.g., identity or log). To estimate the effect of explanatory variables on the share of meat consumption across seven age classes, however, seven separate equations must be estimated. To simplify this technical procedure, we use the generalized structural equation modeling (gsem) command implemented in Stata 16 [41] to fit the proposed model. The gsem command encompasses a broad array of models from linear regression to measurement models to simultaneous equations. In particular, in contrast to the glm command, the gsem command provides a group option, allowing seven equations (representing the seven age classes) being estimated in one step. Therefore, the gsem command is more flexible than the glm command. As one of the major advantages, gsem allows fitted intercepts and parameters to be constrained or to be equal across groups by specifying the ginvariant() option. Note that parameters and standard errors do not differ between the two proposed approaches as long as group-specific intercepts and parameters are not constrained.

To choose the correct distributional family for the gsem command, we conduct the following four tests suggested by Rodriguez [42]:First, we test the basic assumption of a Poisson distribution where the mean and variance are the same. For the two dependent variables, we detect over-dispersion. In the case of (1) share of total meat consumption (in %), the variance (198.3) is 12 times larger than the mean (16.6), and in the case of (2) share of red meat consumption (in %), the variance (118.3) is 14 times larger than the mean (8.7).Second, we estimate a simple Poisson regression and test the goodness of fit (note: age class is included in the equation as an ordinal-scaled variable). The null hypothesis that Poisson is the correctly specified model must be rejected because, for both dependent variables, we obtain a significant *p*-value (0.000) from Pearson’s chi-square test.Third, we estimate a simple negative binomial regression. The corresponding likelihood ratio provides a test of the over-dispersion parameter alpha. For both dependent variables, alpha is statistically significantly different from zero.Fourth, we compare fit indices the Akaike information criterion (AIC), Bayesian information criterion (BIC), and the log-likelihood (LL) for the following two models: Poisson with robust standard errors and Negative binomial with robust standard errors. Overall, the model specification negative binomial with robust standard errors is superior (comparative fit indices AIC, BIC, and LL are presented in Table 3).

Consequently, we choose negative binomial as the distributional family, which allows for over-dispersion. By specifying negative binomial, the log-link function is commonly used as a transformation function. Parameters are estimated by maximum likelihood with robust standard errors. For our variable of interest “gender”, we examine whether coefficients differ across age classes using SUEST (seemingly unrelated estimation) command in combination with the Wald test. The null hypothesis of the Wald test is that the difference between two coefficients is equal to zero. Results of the Wald test can be found in Table A3 in Appendix A.

## 5. Results and Discussion

Table 4 presents the results of the multiple-group regression for the two dependent variables: (1) share of total meat and (2) share of red meat. For the interpretation of model estimates, we report the average marginal effect (AME) of an explanatory variable. The AME indicates the percentage point change in the fractional response variable when the explanatory variable considered is increased by one unit (other explanatory variables are assumed to be held constant).

### 5.1. The Gender Bias in Meat Consumption across Stages of Human Life

Interestingly, the gender difference in meat consumption cannot be observed for infants (Age Class 1, ages 0–4). For both models, the negative effect of gender is statistically not significant. This implies that as long as the gender identity has not developed, the gender bias in meat consumption is non-existent. The gender bias becomes only slightly visible during childhood (Age Class 2, ages 5–11), where boys exhibit a 0.7 percentage point higher share of total meat consumption. Regarding red meat, the gender bias is statistically not significant. Nevertheless, for both, total and red meat, the Wald test indicates a statistically significant difference of the gender coefficient between age class 1 “infancy” and Age Class 2 “childhood” (share total meat *p* = 0.091; share red meat *p* = 0.050). For men, red meat, therefore, seems to gain attractiveness during puberty, when differences in masculinity, femininity, and sex role intensify [43]. While male adolescents show a 1.5 percentage point higher share of total meat consumption, the share of red meat consumption is higher by another 0.2 percentage points. Therefore, for red meat, the difference of the gender coefficient between Age Class 2 “childhood” and Age Class 3 “adolescence” is statistically significantly different from zero (*p* = 0.002). In contrast, for total meat, the difference of the gender coefficient is not statistically significantly different from zero (*p* = 0.316). In early adulthood (Age Class 4, ages 21–35), men attempt to live up to idealized notions of what it is to be a man by demonstrating initiative and assertiveness [44]. Especially for the share of total meat consumption, we observe a steep increase in the gender bias from adolescence to early adulthood. Accordingly, the Wald test reveals that the difference of the gender coefficient between Age Class 3 “adolescence” and Age Class 4 “early adulthood” are statistically significantly different from zero (*p* = 0.020). Men in early adulthood have a 3.2 percentage point higher share of total meat consumption than women. Bruun Eriksen [45] characterizes the subsequent stage, midlife (Age Class 5, ages 36–50), as the crisis of masculinity. On the one hand, men are mandated to become good providers for their families and to be strong and silent. On the other hand, society expects them to take on roles that violate the traditional male code. New skills, such as nurturing children, revealing weakness, and expressing their most intimate feelings, are required. As a result, many men seem to experience a loss of self-esteem and an unnerving sense of uncertainty about what it means to be a man. The gender bias regarding the share of total meat consumption only slightly increases. Consequently, the difference of the gender coefficient between Age Class 4 “early adulthood” and Age Class 5 “midlife” is not statistically significantly different from zero (*p* = 0.581). In contrast, the gender bias regarding the share of red meat consumption slightly decreases. However, the difference of the gender coefficient between Age Class 4 “early adulthood” and Age Class 5 “midlife” is likewise not statistically significantly different from zero (*p* = 0.864). Despite undergoing a partly positive reconstruction of masculinity toward a “loving, enlightened and egalitarian” man in the following stage of mature adulthood (Age Class 6, ages 51–65), traditional attitudes and behaviors that bolster male power are still maintained [46]. The gender bias, therefore, remains at approximately the same level. The Wald test therefore indicates that the difference of the gender coefficient between Age Class 5 “midlife” and Age Class 6 “mature adulthood” is not statistically significantly different from zero (share total meat *p* = 0.677; share red meat *p* = 0.263). Men in mature adulthood exhibit a 3.5 percentage point higher share of total meat consumption than women and a 2.8 percentage point higher share of red meat consumption. Regarding the stage of late adulthood (Age Class 7, ages 66–80), Spector-Mersel [47] (*p*. 73) states that “capitalistic societies do not provide clear final phases for their exalted masculinity stories.” Elderly people crossing the barrier of age 65 are perceived as ungendered and uniform by Western societies. Images of older men especially indicate an inverse correlation between masculinity and aging. Consequently, the gender bias in total and red meat consumption decreases. However, only for red meat, the difference of the gender coefficient between Age Class 6 “mature adulthood” and Age Class 7 “late adulthood” is statistically significantly different from zero (*p* ≤ 0.058). In any case, our hypothesis that the differences in meat consumption between men and women evolves constantly over time has firmly to be rejected.

### 5.2. Further Controls

For the shares of both total meat consumption and red meat consumption, the effect of education of the household reference person is consistently negative across age classes and, in most cases, statistically significant. This implies that the shares of total and red meat consumption decrease with increasing education of the household reference person. For the share of total meat consumption, we observe an increasing effect of education until Age Class 4 (early adulthood) and it remains relatively constant afterwards. For instance, for Age Class 1 (infancy), a unit increase in education decreases the share of total meat consumed by 0.3 percentage points, whereas in Age Class 5 (midlife), a unit increase in education decreases the share of total meat consumed by 1.0 percentage point. For the share of red meat, we identify an increasing effect of education until Age Class 4 (early adulthood) and it decreases constantly afterwards. Findings of other studies on the effect of education on meat consumption point in a similar direction. For instance, Daniel et al. [48] report lower meat intake, particularly red and processed meats, with increasing education. The findings of Zeng et al. [49] indicate that individuals with higher education consume smaller amounts of unprocessed red meat compared to those with lower education.

The effect of household income on the share of total and red meat consumed is consistently negative and statistically significant across all age classes. This implies that the shares of total and red meat consumption decrease with available household income. The age-class-specific (average marginal) effects of household income range between −0.3 and −0.5. Regarding the effect of household income on meat consumption, Guenther et al. [50] provide mixed evidence. While individuals in higher income households consume relatively more chicken, individuals in low-income households consume more processed pork products. Gossard and York [5] find that beef consumption rises with income, whereas income does not affect total meat consumption.

Regarding tendency, larger households consume more meat. Especially for the share of red meat consumption, we mostly observe statistically significant positive effects of household size across age classes. Except for Age Class 3 (Adolescence), however, for both the shares of total and red meat consumed, the effect of household size is statistically significantly negative. Guenther et al.’s [50] findings only indicate a higher probability of beef consumption for larger households (four persons or more) compared to households with two to three persons and one-person households. However, for the other meat types considered, findings suggest non-significant relationships between household size and the probability of meat consumption.

Although men in Western societies usually control family decisions, including the inclusion of meat in meals, meat is often a contested food between marital partners. Food negotiations between partners often conflict about whether, what types, when, and how much meat is consumed [51]. In this context, Eng et al. [52] demonstrate that divorced and widowed men exhibit lower vegetable intake. Therefore, in households where the reference person is married, the shares of total and red meat consumed should be lower. Findings predominantly suggest a negative relationship between marital status of the reference person and the shares of total and red meat consumed across age classes. However, only four class-specific estimators are statistically different from zero at the 10% level. Interestingly, for Age Class 7 (late adulthood), we observe a statistically significant positive effect of marital status of the household reference person on the share of red meat consumed. Persons living in a household where the reference person is married show a 0.8 percentage point higher share of red meat consumption.

Tonsor et al. [53] provide empirical evidence on the relationship between food consumed away from home (eating out) and meat consumption. Their results reveal that increasing consumption of food away from home enhances pork and poultry consumption while reducing beef consumption. Across all age classes, our findings consistently indicate a negative and mostly statistically significant effect of home consumption on the shares of total and red meat consumed. In other words, the higher the share of food consumed at home, the lower the share of meat consumed. For instance, in Age Class 7 (late adulthood), a 10 percentage point increase in at-home consumption decreases the share of total meat consumed by 0.6 percentage points.

## 6. Conclusions

The gender bias in meat consumption has been intensively studied, and the result was unsurprisingly always the same: men eat more meat than women do. However, one piece of the puzzle was missing—namely, the development of a gender bias in total and red meat consumption across stages of human life. By applying a multiple-group regression, we were able to identify a positive and increasing association between meat consumption and evolving masculinity from infancy to late adulthood. In particular, findings reveal for the United States that as long as the gender identity has not developed (ages 0–4), the gender bias in meat consumption is non-existent. The gender bias in the US becomes only slightly visible during childhood (ages 5–11). During the masculinity intensifying stages of human life—namely, adolescence (ages 12–20) and early adulthood (ages 21–35)—the gender bias in meat consumption strongly increases. In the subsequent midlife (ages 36–50) and mature adulthood (ages 51–65) stages of human life, the difference between genders reaches its peak. Crossing the barrier of age 65, there may be an inverse correlation between masculinity and aging. This development might explain why the gender bias decreases during the last stage of human life (ages 65–80).

Meat production and consumption are associated with environmental degradation and growing ethical concerns. Contemporarily, the relationship between (hegemonic) masculinity and meat consumption is challenged by alternative images of masculinity. In this context, vegan and vegetarian diets are gaining increasing popularity among males even though these diets are considered feminine by Western societies. Our findings indicate that the gender bias in meat consumption, at least in the North American context, is non-existent, stagnates, or even decreases as long as the gender identity is not developed. This analysis has many limitations, an important one being the confinement to the socioeconomic environment of the United States. It is certainly not proof of the worldwide primacy of biology in explaining gender differences in meat consumption. Rather, it indicates that biological differences between genders may play a crucial role in the US, a notion that should be followed by in-depth studies on the correlation between biological characteristics and peoples’ preference for meat. A biological link between gender and meat consumption would not making attempts easier to lead men toward nutritional alternatives to meat. Possible pathways will be another rewarding field of future research.

## Figures and Tables

**Figure 1 foods-10-02809-f001:**
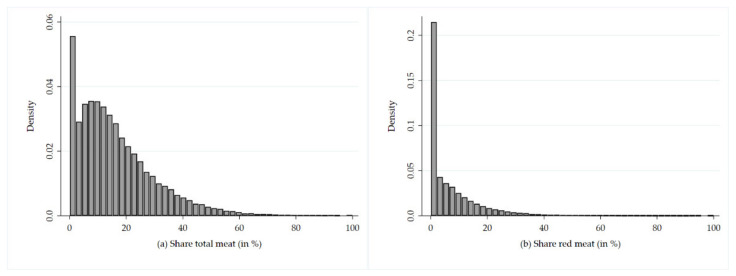
Density distribution of the dependent variables (**a**) share of total meat consumption (in %) and (**b**) share of red meat consumption (in %).

**Table 1 foods-10-02809-t001:** Stages of human life and corresponding age classes.

Stages of Human Life	Description	Age Class	Description
3	Infancy (Ages 0–3)	1	Infancy (Ages 0–4)
4	Early Childhood (Ages 4–6)	2	Childhood (Ages 5–11)
5	Middle Childhood (Ages 7–8)	2	Childhood (Ages 5–11)
6	Late Childhood (Ages 9–11)	2	Childhood (Ages 5–11)
7	Adolescence (Ages 12–20)	3	Adolescence (Ages 12–20)
8	Early Adulthood (Ages 21–35)	4	Early Adulthood (Ages 21–35)
9	Midlife (Ages 36–50)	5	Midlife (Ages 36–50)
10	Mature Adulthood (Ages 51–80)	6 and 7	Mature Adulthood (Ages 51–65) and Late Adulthood (Ages 66–80)
11	Late Adulthood (Age 80+)	7	Top-coded at age 80

**Table 2 foods-10-02809-t002:** Description and summary statistics of the data.

Variables	Description	Mean/ Frequency %	Standard Deviation	Minimum	Maximum
Dependent variables					
(1) Share of total meat consumption (%)	Share of total meat consumption = (total meat consumption in grams/total food consumption in grams excl. beverages) × 100	16.6	14.1	0.0	100.0
(2) Share of red meat consumption (%)	Share of red meat consumption = (red meat consumption in grams/total food consumption in grams excl. beverages) × 100	7.0	10.3	0.0	100.0
Total meat consumption (grams)		337.6	313.,3	0.0	4122.0
Red meat consumption (grams)		142.1	214.6	0.0	3303.9
Total food consumption excl. beverages (grams)		2168.4	1041.6	0.0	14,422.9
**Independent variables**					
Gender (binary)	1 = male; 0 = female	0.5	0.5	0.0	1.0
Age (continuous)	From 0 to max. 80 years	32.4	24.6	0.0	80.0
Education household reference person (ordinal)	1 = less than 9th Grade	10.3			
	2 = 9–11th+ Grade (incl. 12th+ Grade with no diploma)	15.3			
	3 = High school graduate or equivalent	22.8			
	4 = Some college or Associate degree	29.0			
	5 = College graduate or above	22.6			
Household income (ordinal)	1 = under US-Dollar (USD) 20,000	22.7			
	2 = USD 20,000 to USD 44,999	30.7			
	3 = USD 45,000 to USD 64,999	13.6			
	4 ≥ USD 65,000	33.0			
Household size (count)	From 1 person to max. 7 persons	3.8	1.7	1.0	7.0
Married household reference person (binary)	1 = married; 0 = otherwise	0.6	0.5	0.0	1.0
Share home consumption (%)	Share home consumption = (food consumption at home in grams incl. beverages/total food consumption at home and outside in grams incl. beverages) × 100	71.9	26.6	0.0	100.0
Ethnicity (nominal)	1 = Mexican American (base outcome)	18.6			
	2 = Other Hispanic	10.9			
	3 = Non-Hispanic White	37.8			
	4 = Non-Hispanic Black	22.1			
	5 = Other race (incl. multiracial)	10.6			
**Age class**	1 = 0 to 4 years	13.6			
	2 = 5 to 11 years	14.2			
	3 = 12 to 20 years	14.6			
	4 = 21 to 35 years	14.9			
	5 = 36 to 50 years	15.1			
	6 = 51 to 65 years	14.8			
	7 ≥ 65 years	12.8			

**Table 3 foods-10-02809-t003:** Comparative fit indices Akaike information criterion (AIC), Bayesian information criterion (BIC), and log-likelihood (LL) for the two model variants.

Estimation Technique and Model Variant	AIC	BIC	LL
Poisson and (1) share of total meat (in %)	566,054	566,960	−282,922
Poisson and (2) share of red meat (in %)	548,131	549,037	−273,961
Negative binomial and (1) share of total meat (in %)	307,539	308,504	−153,657
Negative binomial and (2) share of red meat (in %)	227,128	228,095	−113,452

**Table 4 foods-10-02809-t004:** Results of the multiple-group regression for the two dependent variables: (1) share of total meat consumption (in %) and (2) share of red meat consumption (in %).

Independent Variables	Share of Total Meat (in %)			Share of Red Meat (in %)		
	*β*	Robust Standard Error	Average Marginal Effect	*β*	Robust Standard Error	Average Marginal Effect
Gender						
Age Class 1	−0.005	(0.030)	−0.0	−0.075	(0.056)	−0.2
Age Class 2	0.056 ***	(0.021)	0.7	0.058	(0.036)	0.3
Age Class 3	0.086 ***	(0.021)	1.5	0.221 ***	(0.036)	1.7
Age Class 4	0.153 ***	(0.020)	3.2	0.291 ***	(0.034)	2.8
Age Class 5	0.168 ***	(0.018)	3.4	0.282 ***	(0.033)	2.5
Age Class 6	0.179 ***	(0.018)	3.5	0.335 ***	(0.034)	2.8
Age Class 7	0.158 ***	(0.021)	2.7	0.239 ***	(0.038)	1.7
**Education of household reference person**						
Age Class 1	−0.048 ***	(0.016)	−0.3	−0.042	(0.029)	−0.1
Age Class 2	−0.025 **	(0.011)	−0.3	−0.024	(0.019)	−0.1
Age Class 3	−0.040 ***	(0.010)	−0.7	−0.075 ***	(0.017)	−0.6
Age Class 4	−0.043 ***	(0.010)	−0.9	−0.090 ***	(0.017)	−0.9
Age Class 5	−0.046 ***	(0.009)	−1.0	−0.079 ***	(0.016)	−0.7
Age Class 6	−0.052 ***	(0.008)	−1.0	−0.071 ***	(0.013)	−0.6
Age Class 7	−0.046 ***	(0.009)	−0.8	−0.074 ***	(0.016)	−0.6
**Household income**						
Age Class 1	−0.038 **	(0.017)	−0.3	−0.125 ***	(0.032)	−0.3
Age Class 2	−0.032 ***	(0.011)	−0.4	−0.078 ***	(0.019)	−0.4
Age Class 3	−0.020 *	(0.011)	−0.4	−0.047 **	(0.019)	−0.4
Age Class 4	−0.023 **	(0.010)	−0.5	−0.055 ***	(0.017)	−0.5
Age Class 5	−0.016 *	(0.009)	−0.3	−0.047 ***	(0.017)	−0.4
Age Class 6	−0.027 ***	(0.009)	−0.5	−0.057 ***	(0.017)	−0.5
Age Class 7	−0.022 **	(0.010)	−0.4	−0.051 **	(0.020)	−0.4
**Household size**						
Age Class 1	0.027 **	(0.012)	0.2	0.083 ***	(0.022)	0.2
Age Class 2	−0.008	(0.008)	−0.1	0.010	(0.014)	0.1
Age Class 3	−0.042 ***	(0.008)	−0.7	−0.051 ***	(0.014)	−0.4
Age Class 4	0.020 ***	(0.007)	0.4	0.024 **	(0.012)	0.2
Age Class 5	0.003	(0.006)	0.1	0.026 **	(0.011)	0.2
Age Class 6	0.001	(0.006)	0.0	0.019 *	(0.011)	0.2
Age Class 7	0.011	(0.009)	0.2	0.001	(0.017)	0.0
**Married household reference person**						
Age Class 1	−0.050	(0.036)	−0.3	0.057	(0.056)	0.1
Age Class 2	−0.004	(0.024)	−0.1	0.026	(0.043)	0.1
Age Class 3	−0.057 **	(0.025)	−1.0	−0.046	(0.042)	−0.4
Age Class 4	−0.056 ***	(0.021)	−1.2	−0.086 **	(0.037)	−0.8
Age Class 5	−0.042*	(0.021)	−0.9	−0.047	(0.038)	−0.4
Age Class 6	−0.013	(0.020)	−0.3	0.014	(0.037)	0.1
Age Class 7	0.036	(0.023)	0.6	0.107 ***	(0.042)	0.8
**Share of home consumption**						
Age Class 1	−0.006 ***	(0.001)	−0.4	−0.009 ***	(0.001)	−0.2
Age Class 2	−0.001**	(0.001)	−0.1	−0.001	(0.001)	−0.0
Age Class 3	−0.002 ***	(0.000)	−0.4	−0.004 ***	(0.001)	−0.3
Age Class 4	−0.002 ***	(0.000)	−0.5	−0.003 ***	(0.001)	−0.3
Age Class 5	−0.002 ***	(0.000)	−0.3	−0.003 ***	(0.001)	−0.3
Age Class 6	−0.002 ***	(0.000)	−0.3	−0.001	(0.001)	−0.1
Age Class 7	−0.003 ***	(0.001)	−0.6	−0.005 ***	(0.001)	−0.3
**Time Fixed Effects**	Yes			Yes		
**Ethnicity Fixed Effects**	Yes			Yes		
**No. of observations**						
Age Class 1	5652			5652		
Age Class 2	5901			5901		
Age Class 3	5960			5960		
Age Class 4	6040			6040		
Age Class 5	6243			6243		
Age Class 6	6115			6115		
Age Class 7	5356			5356		

* *p* ≤ 0.1; ** *p* ≤ 0.05; *** *p* ≤ 0.01.

## Data Availability

The NHANES data set, the only data source for this paper, is publicly available.

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
