# Peer review of "The Old Man and the Meat: On Gender Differences in Meat Consumption across Stages of Human Life"

_foods, 2021, doi:10.3390/foods10112809_

Round 1
Reviewer 1 Report
Even though the topic about meat demand is of interest for sustainable eating, I believe that the current version of the manuscript needs important revision in order to provide a relevant contribute to the literature on determinants for meat consumption and preferences. Below, I provide my detailed comments.
Abstract: description of the results about gender differences across age classes should be less detailed, while the other studied sociodemographic characteristics should be mentioned.
Introduction: in this section, authors should state the research gaps, why the outcomes of their study should be relevant and how they could be used to promote sustainable eating. In particular, they should state why studying gender and age differences is important in analyzing meat consumption.
Age as mediating variable in the gender bias: At the end of the Introduction, the authors write that in this section the hypotheses are presented but a formal statement of them is not included. I suggest the authors to expand the literature about the effect of the other sociodemographic characteristics in a new section and to state hypotheses on each variable of the model.
Data:
- Is the sample representative for gender, age, education and area of residence of US population?
- Lines 173-179: Authors should explain why they compute these 3 variables, especially why they use the first two shares as dependent variables of the models (instead of total (red) meat consumption in grams).
Model specification:
- The model description should start with the explanation of Y (i.e. it represents the share of meat consumption in relation to total food consumed (in %) of respondent i at time (wave) t).
- The descriptions of the βs and ε are missing.
- The specification of gsem is not clear. Authors should provide a more detailed description of the model implementation.
Results and discussion
- Authors should include information about the model fit-
- Lines 298-303: how can you state these findings? By the use of AME? Please, explain this in the text.
- Authors should state if the hypotheses are confirmed by the results of the study. For doing this, they have to perform statistical tests to detect if there are significant differences among the betas of the seven age groups. Gsem should allow this testing in the post estimation tools. The tests should be performed for each variable of the model.
More recent literature should be considered in the discussion of the results.
Conclusions
- The contribution of the study, implications of the results, limitations and suggestions for future research should be included in the conclusions.
Minor:
Tables are not well readable. The title of Table 4 should be corrected (it has two titles)
Reviewer 2 Report
The paper is methodologically well-conceived and realized and it is its strong point, but, in my perspective, it is based on some weak points.
First of all, the database from the National Health 146 and Nutrition Examination Survey (NHANES) of the National Center 147 for Health Statistics (NCHS) of the United States is connotated by the North American culture. Even the Authors consider some cultural factors as relevant, the factors used can be referred to a specific environment and only in some cases replicable in other cultural contexts.
Even if age as mediating variable for gender is acceptable as a hypothesis, some of the mutual affections are not explored considering other variables per gender and per age: what's about the meat consumption of an adult woman unemployed and not married? The marital status or the household size are crossed to age or gender and not with age and gender.
Among the variable considered, a very important factor misses: it's the (new, social, traditional) media influence on consumption. This lack is evident in the Author's analysis of the childhood orientation to eat meat that is referred to parents and not to other important factors as the advertisement is.
Some theoretical assumptions seem dated and debatable. For example, the inverse correlation between masculinity and aging in adulthood proposed around 20 years ago by some scholars needs to be revised in the different societies where we are living, as the idea that midlife means a "crisis of masculinity" as conceived in 1992 by a scholar is different from the current male adults' condition.
Another not well-proved condition is the correlation between the eating of meat and the place where you eat, especially if we consider that these data describe the North American consumer behaviors and that some other factors - for instance the level of income - could favor different food choices out of the home (at the restaurant? on the street? at work?)
These are only some assumptions that are disputable and that make the serious methodological work not complete
Round 2
Reviewer 1 Report
The authors improved the manuscript according to my comments. I appreciated the statistical tests they performed to verify differences in meat consumption among age classes. However, I think that some suggestions that I provided could be better addressed.
- Introduction: authors could better develop this section to underline the importance of studying meat consumption and gender/age differences.
- Model specification: After the model, the authors should explain each symbol. Then, the authors can state how the model is performed: Lines 159-169 and 177-179 should be moved after the explanation of the symbols.
- Conclusions: the limitations of the study should be formally stated, as I previously suggested.
